# Robust 2D Mapping Integrating with 3D Information for the Autonomous Mobile Robot Under Dynamic Environment

**Bin Zhang [1,*] , Masahide Kaneko [2] and Hun-ok Lim [1]**

[1]   Department of Mechanical Engineering, Kanagawa University, Yokohama 221-8686, Japan;
     holim@kanagawa-u.ac.jp
[2]   Department of Mechanical Engineering and Intelligent Systems, The University of Electro-Communications,
     Tokyo 182-8585, Japan; kaneko@ee.uec.ac.jp
[*]   Correspondence: zhangbin@kanagawa-u.ac.jp

**Abstract:** In order to move around automatically, mobile robots usually need to recognize their working environment first. Simultaneous localization and mapping (SLAM) has become an important research field recently, by which the robot can generate a map while moving around. Both two-dimensional (2D) mapping and three-dimensional (3D) mapping methods have been developed greatly with high accuracy. However, 2D maps cannot reflect the space information of the environment and 3D mapping needs long processing time. Moreover, conventional SLAM methods based on grid maps take a long time to delete the moving objects from the map and are hard to delete the potential moving objects. In this paper, a 2D mapping method integrating with 3D information based on immobile area occupied grid maps is proposed. Objects in 3D space are recognized and their space information (e.g., shapes) and properties (moving objects or potential moving objects like people standing still) are projected to the 2D plane for updating the 2D map. By using the immobile area occupied grid map method, recognized still objects are reflected to the map quickly by updating the immobile area occupancy probability with a high coefficient. Meanwhile, recognized moving objects and potential moving objects are not used for updating the map. The unknown objects are reflected to the 2D map with a lower immobile area occupancy probability so that they can be deleted quickly once they are recognized as moving objects or start to move. The effectiveness of our method is proven by experiments of mapping under dynamic indoor environment using a mobile robot.

**Keywords:** SLAM; object recognition; potential moving object detection; immobile area grid map

## 1. Introduction

Recently, autonomous mobile robots have been greatly developed, so that the research field of simultaneous localization and mapping (SLAM) has been noticed, since the almost all kinds of mobile robots need this method for environment understanding, navigation, and path planning [1–3]. Two dimensional (2D) mapping methods based on laser sensors have been widely used, since these kinds of sensors can detect the distances of the obstacles around the robot with a high accuracy stably [4,5]. The moving objects can also be deleted from the map gradually with time going on [4]. However, 2D mapping methods based on laser sensors still have some problems. Three-dimensional (3D) space information cannot be detected by laser sensors. The obstacles can only be detected when they are in the same height with the sensors. Objects with special shapes cannot be reflected to the map correctly, which will cause collisions when the robot moves around according to the map information. For example, a desk with four legs may be detected as four points if the laser sensor on the robot

is set lower than the surface of the desk. The robot might plan its path to go through the desk area and collide with the desk, since there is nothing in this area except these four points in the 2D map. Therefore, 3D mapping methods based on multiple laser sensors or 3D distance sensors have been noticed [6–8]. 3D mapping methods can reflect the shapes of objects in the environment correctly and the robot can plan its path without any collision, but the cost is usually high since view angle of 3D sensors is limited and setting multiple sensors will increase the cost greatly. Although some kinds of cheap 3D distance sensors are developed, they are not stable when moving with a robot and their sensing ranges are limited (e.g., Kinect V2 sensors can only detect the distance information within 8 m with the range of 70 degrees in horizontal angle). Rotating a 2D laser sensor is a good method to detect 3D information, but this method may cause time delay for mapping [9]. Moreover, both conventional 2D and 3D mapping methods takes a long time to delete moving objects from the generated maps, especially when the objects start to move after being sensed as static objects for a while [10–13]. The immobile area grid map method is proposed to solve this problem by adjusting the updating coefficient of moving and static objects [14]. However, they can only delete moving objects or the objects that have started to move and cannot deal with static potential moving objects. For example, static human beings should be deleted from the map, since they will move in the future, but they are usually reflected as obstacles in the map.

Therefore, a robust 2D mapping method integrating with 3D information under dynamic environment is proposed in this paper. The basic 2D map is generated by using a laser sensor, and 3D information of objects, which are detected by an independent 3D sensor, is added to the 2D map by adjusting the occupancy probabilities of different objects. SLAM is then realized by using the immobile area grid map method, in which the immobile probabilities of each grid are updated with different coefficients according to the properties of the objects. Especially, potential moving objects will not be updated to the map if they are recognized. The properties of objects are analyzed from the object recognition results based on RGB-D information from the 3D sensor. The effectiveness of our method is proven by mapping experiments under indoor dynamic environment, where multiple objects and human beings exist. By our method, a robust 2D map containing 3D information can be generated, with the moving and potential moving objects deleted from the map in real time.

## 2. Methodology

The occupancy grid map method is usually used for SLAM, in which the occupancy probability of each grid in the map is calculated and updated. A higher occupancy probability shows that the area is more likely occupied by objects, and this area should be shown as an obstacle in the map. Similarly, a lower occupancy probability shows that the area is seldomly occupied by objects and tends to be an empty area. With time going on, the robot continues sensing the environment, and the occupancy probability is updated. It is a probability model to express each grid. However, this method cannot delete moving objects, since the moving objects are also well detected and will be reflected to the map. Deleting these moving objects case-by-case is still difficult to be applied in complex dynamic environment [15–18]. Generated 2D maps have been widely applied for navigation and path planning for mobile robots, but 2D maps cannot reflect space information, so the robot might collide with the objects (like desks or chairs whose legs only are reflected to the map).

Therefore, we propose to use the immobile area grid map method to conduct SLAM for generating a 2D map containing 3D information. The 3D information is projected to the 2D plane and combined with 2D information before mapping. It is still 2D mapping, so that the processing speed is similar with conventional 2D mapping methods, which is much faster than 3D SLAM. The probability for each grid is calculated by the probability of suitable immobile area instead of the occupancy probability. The probability updating weight is adjusted adaptively according to the properties of the objects. RGB-D information from the 3D sensor is used for object recognition. Recognized objects are used for updating immobile area probability, and recognized moving and potential moving objects are used for updating mobile area probability.

The event that one grid is immobile area is set as *I*, and the event that some objects are observed on the same grid is set as *O*. The immobile area probability $P_t(I)$ can be expressed as Equation (1) at time *t*.

$$P_t(I) \propto P_t(O) \cdot P_{t-1}(I) \tag{1}$$

where $P_t(O)$ is the object existing probability. From Equation (1), the probability $P_t(I)$ that one gird is immobile area at time *t* can be calculated from the observed object existing probability $P_t(O)$ at time *t* and the immobile area probability $P_{t-1}(I)$ for the same grid at time $t-1$. The event *I* can exist, depending on the event *O* can be observed.

Meanwhile, the event that one grid is not immobile area is set as $\bar{I}$, and the event that any objects cannot be observed on the same grid is set as $\overline{O}$. The non-immobile-area probability $P_t(\bar{I})$ can be expressed as Equation (2) at time *t*.

$$P_t(\bar{I}) \propto P_t(\overline{O}) \cdot P_{t-1}(\bar{I}) + P_t(O) \cdot P_{t-1}(\bar{I}) \tag{2}$$

From Equation (2), the event $\bar{I}$ can exist depending on the event *O*, or the event $\overline{O}$ can be observed. It means the sum of the cases that the event $\overline{O}$ (noting is observed happens) and the event *O* (some objects are observed as moving objects) happen. From Equation (2), the following equation can be gotten.

$$P_t(\bar{I}) \propto P_{t-1}(\bar{I}) \tag{3}$$

From Equation (3), the event $\bar{I}$ exists only depending on a constant value. Dividing Equation (1) by Equation (3), Equation (4) can be gotten.

$$\frac{P_t(I)}{P_t(\bar{I})} \propto \lambda \cdot P_t(O) \cdot \frac{P_{t-1}(I)}{P_{t-1}(\bar{I})} \tag{4}$$

where $\lambda$ is the constant coefficient. The updating coefficient for immobile area probability is defined as $\beta$, and Equation (5) can be gotten.

$$\beta = \lambda \cdot P_t(O) \tag{5}$$

where $\lambda$ is defined as the variable parameter that is controlled by the object recognition results. The observed objects are recognized as the immobile areas or moving objects first, and the results are used for setting the value of $\lambda$. Then we can adjust the value of $\lambda$ by the value of $\beta$. The potential moving objects can be deleted for updating the map by adjusting the value of $\lambda$.

The objects around the robot can be recognized. Normally, walls are categorized as still objects, and humans are categorized as moving objects. For the others, all of the objects are treated as still objects until they are moved. It means that the objects are considered moving when they appear or disappear in the environment, comparing with the map generated in the previous frame. Otherwise, they are judged as still objects, even if they are detected only in one frame. The robot does not need to be standstill during the judgment, because the robot can localize itself and the positions of the objects in the environment can be calculated even if the robot moves around. The value of $\lambda$ is adjusted depending on the recognition results. If the results are unknown objects, the object is judged ambiguous and $P_t(O) = 0.5$. The immobile area probability should not be changed, as the observed information is meaningless for updating the map. Thus, the value of $\lambda$ is set as 2 to make sure the updating coefficient for immobile area probability $\beta = 1$. If the recognition results are moving objects, the value of $\lambda$ is set smaller than 2 so that the observed objects are not used for updating the map. If the recognition results are still objects, the value of $\lambda$ is set bigger than 2 so that the observed objects are used for updating the map by a big updating coefficient and the objects can be quickly reflected on the map.

The value of $\lambda$ is set based on the likelihood of the object recognition results. If the likelihood is low, which shows that the object recognition result cannot be trusted, $\lambda$ is set around 2. The higher the likelihood is, the greater the value of $\lambda$ changes. It increases from 2 when recognized as still

objects, like walls or desks, and it decreases from 2 when recognized as moving objects, like human beings. The relationship between object probability and map updating coefficient is calculated by simulation [14], which is shown in Figure 1. In the simulation, we observe that the occupancy probability is updated in a faster speed, with the object probability increasing. However, the immobility probability in the proposed method can be adjusted by changing the coefficient $\lambda$. It means that the updating speed can be faster than conventional methods if the object is recognized to be still, and the updating speed can also be smaller than conventional methods if the object is recognized to be moving. From the simulation result, we chose $\lambda$ as 0.5 when the objects are recognized as human and 10 when the objects are recognized as walls. The flowchart of this process is shown in Figure 2.

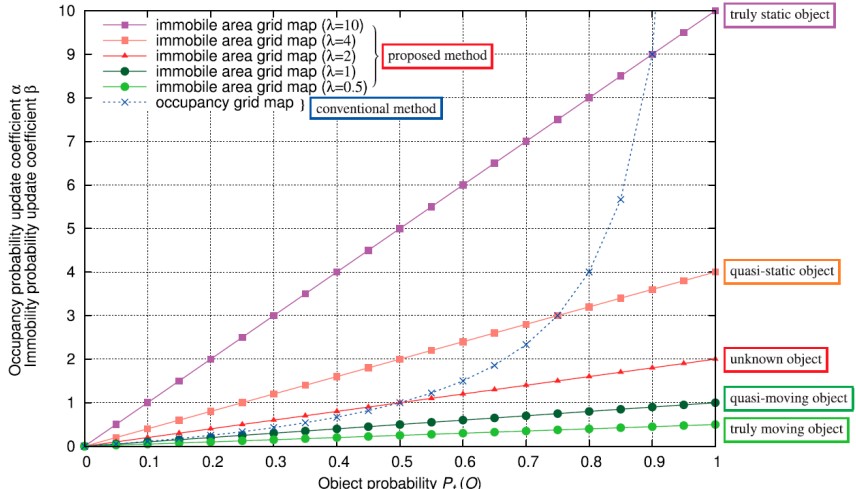

**Figure 1.** Simulation results of the relationships between object probability and updating coefficient [14]. Proposed immobility probability can be updated by different linear equations by adjusting $\lambda$.

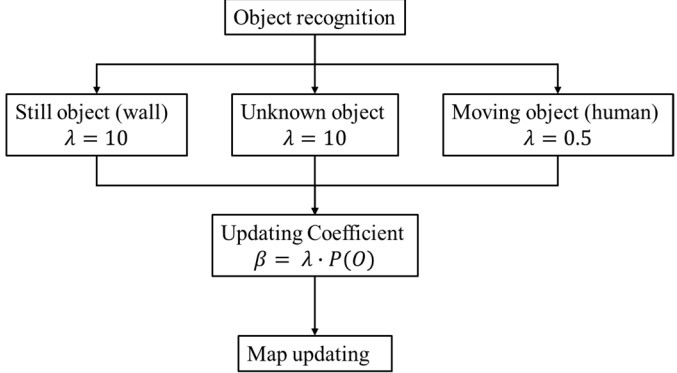

**Figure 2.** Process of setting the map updating coefficient $\lambda$ based on object recognition results. $\lambda$ is set as 10 for still objects, so that immobility probability can be updated faster than conventional method. $\lambda$ is set as 0.5 for moving objects, so that immobility probability can be updated slower.

## 3. Design of the Robot

To realize our method, an autonomous mobile robot equipped with a laser sensor and a 3D RGB-D sensor was developed. The view of the robot is shown in Figure 3a. PIONEER 3-DX made by MobileRobots Co. was chosen as the mobile platform, which is programmable and easy to equip with sensors. LRF sensor was set at the height of 32 cm to scan the 2D distance information for generating the basic 2D map, and Kinect V2 sensor was set at the height of 100 cm from ground (just above the LRF sensor) for getting the RGB-D sensor in front of the robot. Here, Kinect V2 sensor made by Microsoft Co. was used to detect 3D RGB-D information around the robot, since it can sense the color and depth information at the same. It can sense maximum 8 m in front with the range of 70° in horizontal and

60° in vertical. The color information is good enough for object recognition, with the resolution of 1920 × 1080 pixels. It was equipped to the robot horizontally higher than the 2D sensor so that it can detect the obstacles and humans around within its sensing range, as shown in Figure 3b. UTM-30LX made by Hokuyo Co. was chosen as our LRF sensor due to its wide sensing range and high sensing accuracy. It can sense maximum 10 m in front with the range of 270°. The degree step is 0.25°. It was also equipped to the robot horizontally so that it can detect the obstacles and humans around within its sensing range, as shown in Figure 3c.

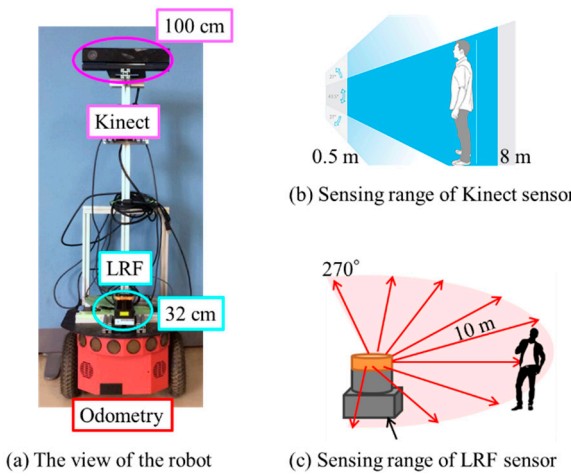

(a) The view of the robot    (b) Sensing range of Kinect sensor    (c) Sensing range of LRF sensor

**Figure 3.** The view of the robot and sensing ranges of different sensors. Kinect sensor is set above LRF sensor so that the origin points are the same in the horizontal coordinate system.

## 4. 2D Mapping Integrating with 3D Information

LRF can detect the distance information in front of the robot in the height of the sensor. As the sensor is set horizontal on the robot, the horizontal distance information can be gotten and used for updating the basic map. For example, the robot starts in the scene shown in Figure 4a, the objects in front of the robot can be detected and reflected to the map as shown in Figure 4b. The position of the robot is shown as the red triangle, with one angle pointing at the robot facing direction. The lines shown in the map present the positions of the obstacles. Kinect V2 can detect the 3D RGB-D information in front of the robot. The objects can be recognized by RGB-D information. For example, the human beings shown in Figure 4a can be recognized by using opensource Kinect SDK [19] and the shape and position information of this person can be gotten from the depth information, as shown in Figure 4c.

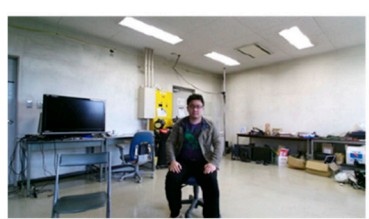 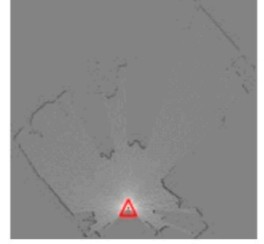 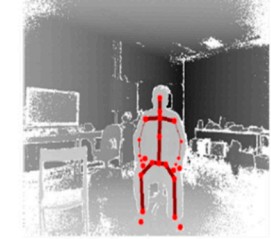

(a) A scene for object detection    (b) Detection result by LRF    (c) Detection result by Kinect

**Figure 4.** Detection results for a scene by different sensors. LRF can get wide-range distance information in a horizontal plane and Kinect can get narrow 3D distance information.

Both the LRF sensor and Kinect sensor are set horizontal against the floor, and they are also set on the same place in the vertical direction so that the projected information from Kinect sensor is overlapped with that from LRF sensor. The planes in the 3D space are detected by using point cloud library (PCL) [20]. For the scene shown in Figure 5a, the planes can be detected (Figure 5b).

As the robot is working in the indoor environment, we segment the highest horizontal plane with the threshold higher than 2 m as the ceiling plane (Figure 5d). The vertical planes connected to the ceiling plane are recognized as the walls (Figure 5e). The horizontal plane in the height of the robot wheel is recognized as the floor (Figure 5c). The surfaces of other objects like desks can be also recognized as planes. As shown in Figure 6a, the desk is detected as four points by LRF, since only the four legs of the desk are detected (Figure 6c). The surface of the desk is detected by the plane detection process, and they are shown as the blue area in Figure 6b, after projecting on the ground. Besides, the red dots show the wall information (the points that are recognized as a wall in 3D information gotten by Kinect). The blue dots show the plane information of other objects (the points that are recognized as planes in 3D information gotten by Kinect). The black dots show the areas that are occupied by objects in the map and white dots show the areas that are not occupied. The gray dots show the areas that are still unknown. The cyan dots show the laser information gotten from the LRF sensor. The angles of these two sensors were adjusted to horizontal against the ground before the experiments. To solve the problem of vibration, the 3D information used for projection was corrected by using the estimated angle of the ceiling planes. As for the LRF sensor, it was set at the height of 32 cm over the ground and was used to get 2D information to generate the basic map. The sensor was fixed to the robot base at a low height so that the vibration was small. Meanwhile, the vibration for a few frames did not influence the mapping result, since the sensing results were updated online every once in a while, and calculated as probability.

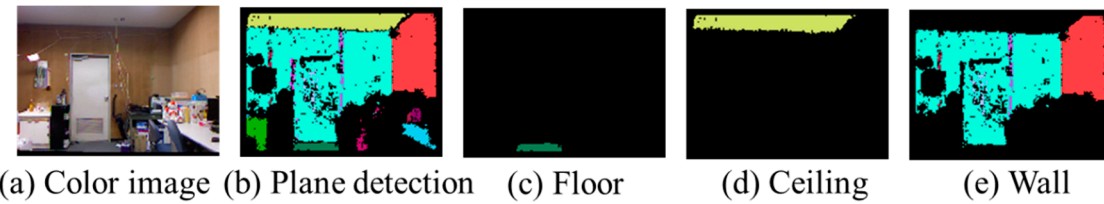

**Figure 5.** Plane detection results and the detection results of the ceiling, floor, and walls.

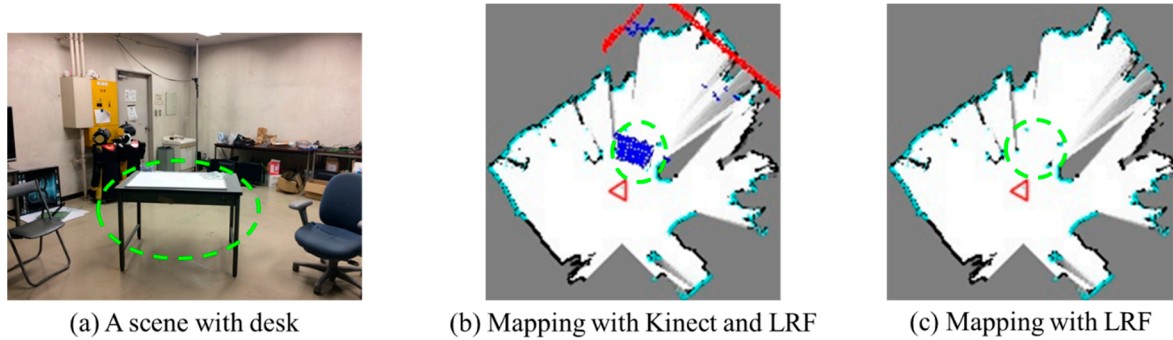

**Figure 6.** Desk detection results by different methods. It is detected as four points by the LRF sensor, while it can be detected as an area by combing two kinds of sensors.

In our system, the LRF sensor is always used during the mapping process. This sensor is used to build the basic 2D map, since the sensing range and the sensing distance of the LRF sensor are all bigger than the Kinect sensor. The basic 2D map can be built fast by using the LRF sensor. However, the LRF sensor cannot detect 3D information. The Kinect sensor is used to recognize 3D objects and reflect the recognition results to the 2D map. Since the data amount from the Kinect sensor is bigger, the recognizing and projecting processes are slower than the LRF sensor, so that the Kinect information is not continuous but only useful in a few frames. By combing the information of these two sensors, the robot can map at a fast speed and generate a robust map with 3D information. They are adjusted to be in the same place in the vertical direction so that when the 3D information is projected to the ground, the coordinates of the 2D and 3D sensors are overlapped. The accuracy of the LRF and Kinect

sensors are similar, so that the error between them can be ignored, especially in the grid map where one grid point expresses the distance of 50 mm. The 3D surface information of the desk is not completely overlapped with the four legs in 2D information in Figure 6b, because the plane recognition result of desk surface by the Kinect sensor is not good enough. This is caused by reflection, as well as the plane detection accuracy of PCL. This problem can be solved easily just by moving the robot until the whole desk surface area is detected. The 3D information is projected to the ground, integrating with 2D information, and the integrating result is shown in Figure 7. The white areas show empty areas, red lines show wall, blue areas and dots show projected objects, and light blue lines show the objects detected by LRF. It is observed that 3D information is reflected on the map correctly.

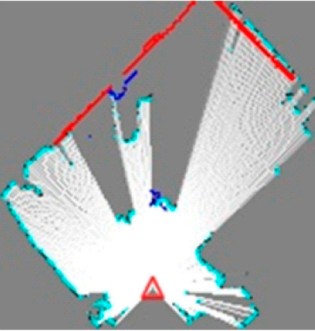

**Figure 7.** Mapping result after integrating two kinds of information. Kinect information (red and blue areas) enriched the 2D map generated by using LRF information (green area).

## 5. Experimental Results and Analysis

The effectiveness of our method was proven by mapping experiments under indoor dynamic environment, where multiple objects and human beings existed. The experimental environment is shown in Figure 8. The robot was controlled manually by the user when moving around the environment. It moved around in the room and went out to move in the passage, shown as the dark green line. The two people (Figure 8d, circled by the green line) in the room kept still during the experiment. The person near the door moved out of the room to open the door for the robot (Figure 8e,f, circled by the green line). The person in the passage kept moving in the beginning and stopped after a while (stopped at Figure 8g, circled by the green line). There were multiple obstacles in the room and the passage. The obstacles in the corner near the back door (Figure 8a, circled by the red line) were some boxes put behind the screen. Some other boxes were put in another corner of the room (Figure 8b, circled by the red line). The frontal door (Figure 8c, circled by the red line) was opened by a person to help the robot move out of the room. In order to compare our results to that based on the conventional method, the user controlled the robot to move at the same speed, and the maps were generated in the same environment. The information processing speed was around 10 frames per second. The mapping result by the proposed method is shown in Figure 9a, and the mapping result by the conventional occupied grid map method is shown in Figure 9b. These maps were generated in around 5 min.

From the results, it is observed that all the humans were well detected and deleted from the map in the proposed method ((d–g), circled by the green line in Figure 9a), whether they were static or moving. On the other hand, in the conventional method, the humans who were static during the experiment were not able to be deleted in the map ((d), circled by the green line in Figure 9b). As for the moving people, some were deleted from the map slower than the proposed map ((e), circled by the green line in Figure 9b) and some appeared in the map ((f,g), circled by the green line in Figure 9b). The differences were caused by the updating method when generating the map. Since the humans were recognized by the proposed method, the immobility probabilities of the human areas were updated slowly by adjusting the updating coefficient $\lambda$ as 0.5. As for the occupancy probabilities in the conventional method, they were updated fast, since the static human areas were also kept observed as objects so that they ((d), circled by the green line in Figure 9b) were left in the map. The person near

the door ((e), circled by the green line in Figure 9b) was shown in the map for the same reason. Even when this person started to move, it took time to update the occupancy probability so that it was not completed deleted. Only the persons who kept moving were deleted for the conventional method (no human moving trajectories in the map). Besides, the persons may appear in the map easily if they stop, since they will be observed as static objects continuously ((f,g), circled by the green line in Figure 9b). The results prove the effectiveness of our method, which can delete moving and potential moving objects in real time when generating a 2D map.

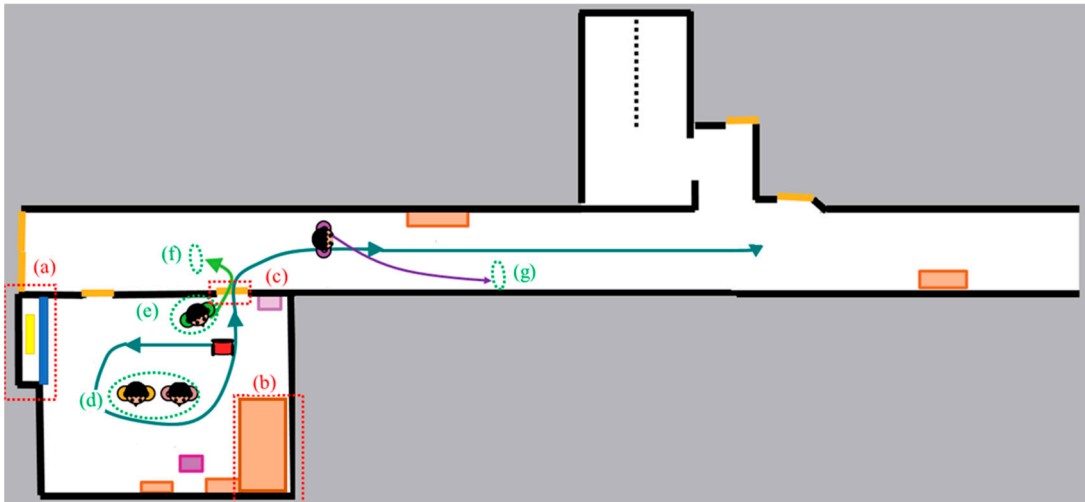

**Figure 8.** The experimental environment with multiple objects and humans. The robot moved around in the room and went out to move in the passage, shown as the dark green line.

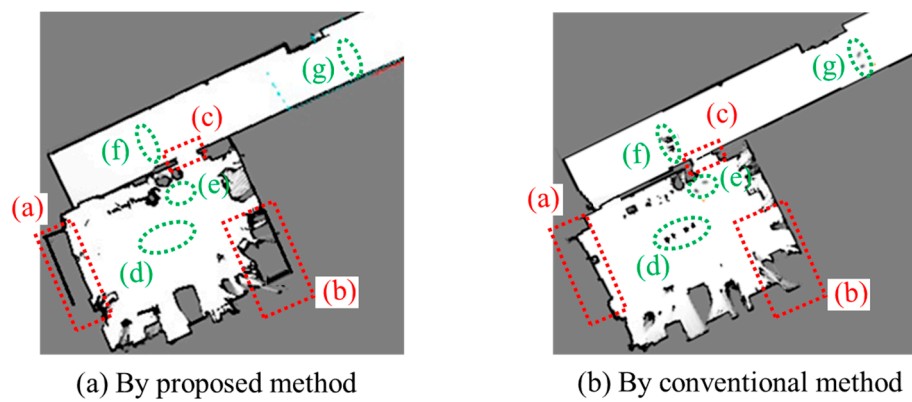

(a) By proposed method　　　　　　　　　(b) By conventional method

**Figure 9.** Mapping results of the lab environment by the proposed (**a**) and conventional (**b**) methods. The map generated by the proposed method contains more information of the environment and deleted moving and potential moving objects quickly from the map.

Besides, we observe that the room area generated by the proposed method ((a,b), circled by the red line in Figure 9a) is quite different from that generated by conventional method ((a,b), circled by the red line in Figure 9b). The room is bigger when using the proposed method, which better expressed the area of the room by showing its outline correctly comparing with Figure 8. This difference is caused by the 3D information. The occluded areas in the room which were blocked by some screens or boxes on the ground were also detected in the proposed method by detecting the upper part of the walls, so that the map area is bigger and clearer than the map generated by the conventional method. Meanwhile, we observe that the frontal door area was deleted in the map generated by the proposed method ((c), circled by the red line in Figure 9a), but it was left as a line on the map generated by the conventional method ((c), circled by red line in Figure 9b). The door was not recognized as a moving or potential

moving object in the beginning, so it was reflected on the map generated by both methods. However, the door area was deleted quickly after being opened by the person near the door by the proposed method, since the immobile probability was updated quickly. On the other hand, the door area was deleted slowly by the conventional method, since the occupancy probability was updated slowly. As a result, the door was not deleted completely before the robot moved away (the map cannot be updated if the area is not observed by the sensors of the robot) and was still left on the map generated by the conventional method. The results prove the effectiveness of our method, generating a 2D map containing 3D information. This map is more robust for path planning of mobile robots.

One of our experimental videos has been uploaded to the Internet: *https://youtu.be/es-xEQYyBYI*.

## 6. Discussion

Mapping is one of the key technologies for autonomous mobile robots, since it is the foundation of path planning and motion control. Robots tend to move safely and efficiently if the map is more accurate and able to deal with dynamic environment. 2D maps generated by open sources like gmapping [21] are widely used for many kinds of service robots nowadays, because 3D mapping methods usually need more processing time for saving and updating the space information [6–8]. Motion planning in 3D maps also needs more processing time, and the algorithms are usually complex so that they are not widely used by general service robot developers. However, 2D mapping methods usually cannot reflect the environment sufficiently [14,22,23]. The shapes of objects and the occluded areas are always ignored, which may cause dangers or miss the best route for path planning. A robust 2D mapping method containing 3D space information is proposed in this paper to solve these problems. By combing 3D space information with 2D distance information, the generated 2D map can reflect the environment sufficiently, which is generated quickly, with a similar speed to conventional 2D mapping methods. We observe that our method can reflect the spatial shape information of objects correctly in the 2D map from the experimental results. The spaces occluded by objects can also be detected and reflected to the 2D map.

The accuracy of the generated 2D map by the proposed method has also been improved greatly. In the proposed method, the immobile area grid map is used for expressing the environment, and object recognition results are used for adjusting the coefficient to update the map. In this way, the recognized moving and potential moving objects (e.g., moving or static human beings) will be deleted from the map, since this information will not be used for updating the map from the beginning. Meanwhile, the changes of the environment (e.g., the door is opened during the mapping process) will be reflected to the map quickly compared with conventional mapping methods based on occupancy grid maps [15–18,23]. The generated 2D map can be applied for controlling the robot in the same ways as before. Meanwhile, it can improve the accuracy of path planning and the safety when controlling the mobile robots.

On the other hand, although the proposed method shows great performance in deleting moving and potential moving objects from the map quickly, the rationality to express all the objects in this way still needs to be discussed. For example, human beings need to be deleted from maps, but the doors, which are opened during the mapping process, are also deleted by our method. It is better than leaving part of the door (needs time to be completed deleted) in the map compared with the conventional methods based on occupancy grid map, but the doors may be closed again in the future. Leaving them as movable objects in the map may be a better way to express the environment. Moreover, there is still a limitation of 3D information reflected in the 2D map. The robot can avoid colliding with the objects by the proposed method, but the 3D information is not enough for the robot to manipulate the objects (e.g., taking the objects by a robot hand). Furthermore, the experiments were conducted in our laboratory, where all the objects and their motions could be totally controlled. Experiments with more variable and unpredictable conditions under real world environments need to be conducted before applying this method to real service robots.

## 7. Conclusions

In this paper, a robust 2D mapping method integrating with 3D information under dynamic environment was proposed. 2D distance information gotten from LRF was used for generating the basic 2D map, and 3D RGB-D information was added to improve the accuracy of the map. Objects were detected and recognized by RGB information and their top view shapes are reflected to the 2D map. The updating coefficient was adjusted according to the recognition results of the objects, where moving and potential moving objects were not used for updating the map. SLAM was realized by using the immobile area grid map method, in which the immobile area occupancy probability of each grid was updated according to the properties of the objects. The moving and potential moving objects were deleted from the map in real time, and shapes like chairs and desks were reflected to the map as areas rather than some points. The effectiveness was proven by experiments conducted under indoor dynamic environment.

In the future, we will continue working on recognizing and analyzing the properties of different objects and show the properties of objects on the map. For example, the robot can stand near the wall, but it should not stay behind a door. By this method, the robot can move more naturally like real human beings.

**Author Contributions:** Both B.Z. and M.K. participated in designing the robot, proposing the mapping algorithm, and conducting experiments. H.-o.L. gave important suggestions to submit and revise the paper. All authors gave final approval for publication.

**Funding:** Part of this research was funded by the Shotoku Science Foundation.

**Conflicts of Interest:** The authors declare no conflicts of interest.

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
