# Peer review of "Robust 2D Mapping Integrating with 3D Information for the Autonomous Mobile Robot Under Dynamic Environment"

_electronics, doi:10.3390/electronics8121503_

Round 1

Reviewer 1 Report

1. The article has a structure different  from IMRAD but the overall organization of the paper is satisfactory. The introduction state the purpose of the paper.The conclusions are appropriate.The references are modern.

2. Matching the article to the subject of the journal is good. The topic is important to colleagues working in the field of robotics. The article is interesting and valuable.

3. The article is intelectually stimulant.

You should edit the article and put some text below figure 5.

Author Response

1.Thank you for your advice. The article is modified to the structure of IMRAD to make it easier to be read. 

2. Thank your for your comments. This work is in the area of systems & control engineering, and provide a new method for mapping.

3. The experimental results have been improved.

The article has been re-edited and there are texts below figure 5 and all the other figures.

Please check the revised paper for the corrections in detail.

Reviewer 2 Report

In the paper some complex mechanisms allowing to recognize the dynamic environments for robot usefulness are proposed. After combining the behaviors of 3D and 2D approaches, a new method has been created and successfully confirmed under study. In my opinion the manuscript is well written and organized. I have only two crucial doubts. At the beginning, I am confused since the paper has poor mathematical background not related to such high impact factor journal as Electronics. Secondly, the main topic of the paper is not in the scope of Electronics, probably. The solution of these statements should be provided by Editor of the manuscript, in my opinion.

Author Response

1. The the proposed method in this paper is based on Bayes' theorem and it is a a sequential Bayesian framework. Occupancy grid map is usually used for mapping. The environment is dived into several girds in the map and each grid is expressed by the occupancy probability to show whether it is occupied by objects or not. The map is updated by finding the maximum-a-posteriori (MAP) solution of the joint probability. Our method improved this method by expressing the map as the immobile area occupancy grid map. The immobility probability calculated from the Bayesian framework is adjusted by the object recognition results. The mathematical background in this paper is sufficient, and visualization of equation deducing of Bayesian framework is complex. It has been proven effective in advanced researches [14]. In this paper, we showed the way to improve from the conventional methods and using proportional form to make the process easier to understand. The methodology part is showed in the paper as section 2 in the revised paper. 

2. This work is in the area of systems & control engineering, one of the scope of Electronics, and provide a new method for Simultaneous Localization and Mapping (SLAM). When controlling mobile robots, SLAM is the first step for understanding the environment around the robot. The system proposed in the paper can help the robot generate a more robust map and provide more accurate information for path planning motion control of the robots. A discussion part has been added to the paper to show how this work connects with the broader work of the community.

Please check the revised paper for the corrections in detail.

Thank you.

Reviewer 3 Report

The paper proposes the use of 2D sensor (laser range finder) with 3D sensor (kinect) for simultaneous localization and mapping (SLAM). The claim made in the paper is that such a method is faster and more accurate than individual sensors. This appears to be the novel part of the paper.

A good methods section that explains what has been done is missing. There is bits and pieces of information Fig 3 and 4 but thats it. The paper needs more equations and flowcharts and perhaps tables to show the approach.

Fig 11 and 12. What is the conventional method in Fig 12. I expected to see similar things in the two figures so that one can compare and see which method is better.

The results section (Sec 5) is short. The results are briefly documented. The claim that the method saves time has not been shown in the paper but seems important to support the claim made in the abstract. 

A discussion section that shows how this work connects with the broader work of the community is beneficial.

There a lot of grammar and sentence structuring issues and should be improved. The entire paper needs to be re-edited. For example, abstract this sentence is too long: By using immobile area occupied grid map method, recognized still .. moving objects or start to move. Another example is the introduction. While it contains good information it is not written using correct grammar: Especially, researchers .. (need reframing); 2D mapping methods based... (two sentences are joined awkwardly. I am omitting other parts here but the entire paper needs to be re-edited.

The figure captions should be more descriptive instead of being one-liners. Some of the figures can be combined. For example, 1, 2, and 3 can be combined. Figures 4 and 5 can be combined and so on.

Author Response

1.The methodology part is showed as section 2 in the revised paper. The system proposed in the paper can help the robot generate a more robust map and provide more accurate information for path planning motion control of the robots. The map contains more information of the space, moving or potential moving objects can be deleted from the map, and the map can be updated quickly once the environment changes.  

2. Figure 8 and 9 are added to show the differences between the maps generated by proposed method and conventional method. The explanations are also added to the experimental results and analysis part in the revised paper.

3. The experimental results and analysis part in the revised paper is re-edited totally to show the experimental results in detail. The proposed fast 2D mapping method means that our method is still generating 2D method, and it is similar processing speed with conventional 2D mapping methods. But it contains 3D space information. 2D mapping methods are usually faster than 3D mapping because it takes a lot of time for saving and updating all of the points in the space for expressing 3D information when using 3D mapping methods.

4.A discussion part has been added to the paper to show how this work connects with the broader work of the community.

5.The grammar and sentence structuring issues have been improved. The whole paper has been re-edited. The edited parts are shown in red color. 

6.The figure captions are edited to be more descriptive. Some figures are combined. Figure 1, 2, and 3 are combined as Fig. 1. Figure 4 and 5 are combined as Fig. 2. Figure 11 and 12 are combined as Fig.8.

Please check the revised paper for the corrections in detail.

Thank you.

Round 2

Reviewer 3 Report

The Discussion section is too short. It can be improved substantially. The discussion should bring out the major/minor findings in a broader context. It might also go back to the literature and inform the readers about how this work compares (supports/does not support) work done by other researchers. 

I suggest that the authors read the discussion section of other good researchers to get an idea on what the discussion section could entail

Author Response

Thank you for your suggestion. The discussion part has been improved to bring out the findings in a broader context and to compare with conventional works after referring some papers.

The paper has also been re-edited to improve the English language and style.

The modifications in detail can be found in the revised paper.